# Biomimetic Nanoparticles for Basic Drug Delivery

**DOI:** 10.3390/pharmaceutics16101306

**Published:** 2024-10-07

**Authors:** Andrey Tikhonov, Artyom Kachanov, Alexandra Yudaeva, Oleg Danilik, Natalia Ponomareva, Ivan Karandashov, Anastasiya Kostyusheva, Andrey A. Zamyatnin, Alessandro Parodi, Vladimir Chulanov, Sergey Brezgin, Dmitry Kostyushev

**Affiliations:** 1Laboratory of Genetic Technologies, Martsinovsky Institute of Medical Parasitology, Tropical and Vector-Borne Diseases, First Moscow State Medical University (Sechenov University), 119991 Moscow, Russia; tikhonov_a_s@staff.sechenov.ru (A.T.); kachanov_a_v@staff.sechenov.ru (A.K.); yudaeva_a_d@staff.sechenov.ru (A.Y.); ponomareva.n.i13@yandex.ru (N.P.); ivan.karandashov@gmail.com (I.K.); kostyusheva_a_p@staff.sechenov.ru (A.K.); brezgin_s_a@staff.sechenov.ru (S.B.); 2Department of Pharmaceutical and Toxicological Chemistry, First Moscow State Medical University (Sechenov University), 119146 Moscow, Russia; danilik_o_n@student.sechenov.ru; 3Division of Biotechnology, Sirius University of Science and Technology, 354340 Sochi, Russia; parodi.a@talantiuspeh.ru; 4Faculty of Bioengineering and Bioinformatics, Lomonosov Moscow State University, 119234 Moscow, Russia; zamyat@belozersky.msu.ru; 5Belozersky Institute of Physico-Chemical Biology, Lomonosov Moscow State University, 119992 Moscow, Russia; 6Department of Infectious Diseases, First Moscow State Medical University (Sechenov University), 119991 Moscow, Russia; vladimir@chulanov.ru

**Keywords:** RNA, cargo, therapeutics, targeted delivery, delivery vehicles

## Abstract

Biomimetic nanoparticles (BMNPs) are innovative nanovehicles that replicate the properties of naturally occurring extracellular vesicles, facilitating highly efficient drug delivery across biological barriers to target organs and tissues while ensuring maximal biocompatibility and minimal-to-no toxicity. BMNPs can be utilized for the delivery of therapeutic payloads and for imparting novel properties to other nanotechnologies based on organic and inorganic materials. The application of specifically modified biological membranes for coating organic and inorganic nanoparticles has the potential to enhance their therapeutic efficacy and biocompatibility, presenting a promising pathway for the advancement of drug delivery technologies. This manuscript is grounded in the fundamentals of biomimetic technologies, offering a comprehensive overview and analytical perspective on the preparation and functionalization of BMNPs, which include cell membrane-coated nanoparticles (CMCNPs), artificial cell-derived vesicles (ACDVs), and fully synthetic vesicles (fSVs). This review examines both “top-down” and “bottom-up” approaches for nanoparticle preparation, with a particular focus on techniques such as cell membrane coating, cargo loading, and microfluidic fabrication. Additionally, it addresses the technological challenges and potential solutions associated with the large-scale production and clinical application of BMNPs and related technologies.

## 1. Introduction

In recent years, the field of drug delivery has witnessed significant advancements, particularly with the development of nanoparticle (NP)-based systems. Among these, biomimetic nanoparticles (BMNPs) have garnered considerable interest due to their potential to combine the biocompatibility of natural extracellular vesicles (EVs) with the versatility and manufacturability of synthetic NPs. EVs refer to a broad category of biologically secreted nanoparticles, including exosomes, apoptotic bodies, microvesicles, and others [1]. These NPs serve as natural carriers of biomolecules, such as nucleic acids, proteins, and lipids, playing crucial roles in cell–cell communication, disease etiology, and progression, as well as serving as promising delivery vehicles for therapeutic cargo. This is particularly important given the unique structure of EV membranes, which defines their biocompatibility, safety, ability to traverse biological barriers, and programmability [2,3]. The surface programming of EVs can be achieved through various approaches, including genetic engineering, chemical modification, or the incorporation of anchoring molecules into the membranes [4]. The functionalization of EV surfaces has enabled efficient delivery to challenging targets, such as the brain. However, the clinical translation of EVs faces several challenges, including low yield, high heterogeneity, and complex standardization processes [5]. Mesenchymal stem cells (MSCs) are considered one of the safest sources for obtaining extracellular vesicles (EVs). However, their standardization is inadequate, with significant variations arising from the source of isolation and the characteristics of the donor [6]. The most common method for obtaining EVs is the differential ultracentrifugation of conditioned media [7]. This method presents challenges related to scalability and the maintenance of the final product’s purity, which limits its suitability for industrial applications. Furthermore, the cargo packaging methods employed in EVs allow for the effective and efficient encapsulation of proteins and nucleic acids. However, technical challenges remain in the packaging of virus-like particles, nanoscale particles such as upconversion nanoparticles, and ribonucleoprotein (RNP) complexes like CRISPR-Cas9 [8,9].

To address these limitations, BMNPs that can mimic the structural and functional characteristics of EVs have been developed. The techniques employed in the production of BMNPs yield a more homogeneous and purified product. Furthermore, the internalization efficiency and biological activity of these nanoparticles are superior to those of EVs when administered at equivalent doses. Based on the preparation methods, BMNPs can be classified into three main types as follows: (1) cell membrane-coated nanoparticles (CMCNPs), (2) artificial cell-derived vesicles (ACDVs), and (3) fully synthetic vesicles (fSVs). Each of these types has distinct advantages and faces challenges in development and application.

While the potential of BMNPs is considerable, a few technological challenges must be overcome before their full clinical applications can be implemented. First, it important to guarantee the integrity and functionality of the biomimetic coatings to optimize the efficiency of cargo loading and to standardize production processes in the development of BMNPs. Moreover, improvements must be made to the scalability and cost-effectiveness of BMNP production to facilitate their widespread use.

Here, we provide a comprehensive review of the current state of BMNP research, with a particular focus on the preparation methods, applications, and challenges associated with CMCNPs, ACDVs, and fSVs. By reviewing recent advances and identifying key areas for future research, we aim to contribute to the development of next-generation drug delivery platforms that can bridge the gap between the laboratory and the clinic.

## 2. Preparation of Biomimetic Nanoparticles and Cargo Loading

Considering the limitations of EV exploitation in targeted delivery approaches, the preparation of BMNPs that mimic properties of natural EVs but are more convenient and standardizable during manufacturing is a potential approach for biomedical applications.

BMNPs can be prepared using top-down or bottom-up strategies (Figure 1). In the top-down approach, bulk materials are broken down into smaller particles that can be subsequently used for drug delivery. The most frequently used top-down strategy employs the coating of NPs with cell membranes. In this approach, cell membranes are isolated and attached to smaller carriers camouflaging their synthetic surface [10]. Another example of a top-down approach is a method in which whole cells are disintegrated and used as precursor material to generate smaller nanovesicles [11] called exosome-mimetic nanovesicles or cell-derived nanovesicles. However, according to the minimal information from studies of extracellular vesicles (MISEV2023), the term artificial cell-derived vesicles (ACDVs) for vesicles generated from extruded cells is recommended [12]. ACDVs can be produced by using virtually any cell type, saving up to 3 days of manufacturing, and providing ~250-fold more product than naturally derived EVs [13,14]. Alternatively, a bottom-up strategy utilizes basic components, such as proteins and lipids, as building blocks to form nano-sized biomimetic carriers. The bottom-up approach allows for the full control of all particle characteristics and provides greater potential for scalability.

## 3. Top-Down Approaches

### 3.1. Cell Membrane-Coated Nanoparticles (CMCNPs)

The most frequently used top-down strategy involves coating NPs with cell membranes. In this approach, cell membranes are isolated and attached to the NP surface to camouflage the NP’s synthetic components [10]. The fused cell membrane preserves its natural characteristics and facilitates the evasion of host immunity and phagocytosis by mimicking natural cells or other biological objects. Such coatings can extend particle bloodstream circulation time, inhibit recognition and clearance by macrophages of the reticuloendothelial system (RES), and decrease non-specific cytotoxicity of synthetic NPs [15].

The synthetic core of NPs can be cloaked with a variety of different biomimetic coatings, such as the membranes of immune cells (leukocytes [16], neutrophils [17], T-cells, NK cells [18], macrophages [19]), platelets (PLTs) [20], tumor cells [21], red blood cells (RBCs) [22], and mesenchymal stem cells (MSCs)) [23]. Every coating has its own advantages and disadvantages. For instance, coating NPs with macrophage membranes promotes inflammation targeting and helps NPs evade the mononuclear phagocyte system; erythrocyte membranes inhibit the phagocytosis of NPs and prolong circulation time; and MSC membranes have a natural affinity toward cancer cells and areas of inflammation [24]. Besides cell membranes, EV membranes [25] and organelle membranes (e.g., mitochondrial membranes [26]) can be utilized to coat NPs as well.

The process of preparing cell membrane-coated nanoparticles (CMCNPs) consists of the following three steps: (1) cell lysis, (2) extraction of the cell membrane, and (3) coating NPs with the cell membrane. These procedures should be as gentle as possible to preserve functionality of the molecules on the membrane surface.

#### 3.1.1. Cell Lysis

Numerous methods can be utilized for cell lysis, such as hypotonic lysis, homogenization, sonication, freeze–thaw, or nitrogen cavitation. Hypotonic treatment is often used in preparing CMCNPs; a hypotonic solution is added to cells to induce osmotic swelling and subsequent cell lysis due to osmotic pressure imbalance. Hypotonic buffers are inexpensive, and the procedure is simple and not labor-intensive, as well as mild and non-destructive to membrane proteins. However, it has low fragmentation efficiency and may alter the intact structure of membrane fragments [27].

A Dounce homogenizer is often used to mechanically lyse cells by the shear force created when the sample is forced through the device’s tip. Homogenization is efficient but requires specialized equipment and can damage cell membranes [27].

Sonication is another method for cell lysis. Samples are subjected to ultrasound energy, which causes cavitation in the surrounding liquid and subsequent membrane disruption. Sonication is an easy-to-use and effective technique with low equipment requirements. However, membranes can be affected by the free radicals generated during sonication, and the released energy can heat samples, causing protein denaturation [28].

The freeze–thaw technique employs several freeze–thaw cycles to disrupt cells. The formation of ice crystals during freezing and their subsequent melting ruptures cells. Freeze–thaw is a relatively uncomplicated method, but can cause protein damage [27,29].

Nitrogen cavitation takes place in a nitrogen cavitation chamber and involves nitrogen bubble formation. When pressure is applied to the sample, nitrogen diffuses into the cells. When the pressure drops, nitrogen forms bubbles inside the cells, generating enough force to break them. Nitrogen cavitation is time-consuming, requires expensive equipment, and may change cell membrane morphology, so it is rarely used for cell lysis.

It is suggested that the preferred lysis technique varies with cell type. For example, homogenization is recommended for larger cells, while hypotonic treatment or freeze–thaw is advised for RBCs and PLTs, since these techniques do not require complicated operation [30]. However, there is a clear need for studying the efficacy, universal applicability and damage of membranes by cell lysis methods. In particular, their impact on membrane charge, cell targeting properties, and the integrity of cell membranes and receptors/ligands on the surface, as well as the activity of complex biomolecules, which is currently not clear. Advanced methods for mild but efficient cell lysis and the subsequent isolation of membranes are required.

#### 3.1.2. Membrane Extraction

The second step is membrane extraction, which is usually performed by density gradient ultracentrifugation (DGU), ultracentrifugation (UC), or medium-speed centrifugation. DGU is often used to derive cell membranes from subcellular components and soluble proteins. The drawbacks of DGU-based membrane isolation are its time-consuming nature and specialized equipment requirements. Other membrane extraction alternatives are medium-speed centrifugation and UC, but these methods yield lower purity and specificity of the obtained cell membranes [31]. After the centrifugation step, a pellet containing cell membranes is obtained. If purified cell membranes are not immediately needed for further steps, they can be kept at −80 °C [32]. Cell membrane isolation from anucleate cells is relatively uncomplicated, with a single extraction step. However, for cells containing a nucleus, the process is more complex and laborious, requiring multiple purification and isolation steps [33] to separate cell membranes from intracellular organelles and proteins.

Commercial kits are available for cell membrane extraction, such as the Plasma Membrane Isolation Kit and the Minute™ kit. These kits rely either on DGU to isolate membranes after adding the appropriate lysis buffer, or on the use of a spin column-based approach. However, commercial kits are relatively expensive to use, not suitable for large-scale production, and may still require the use of an ultracentrifuge.

Cell membranes can be modified after isolation to endow them with new properties. For instance, hyaluronidase-modified RBC membranes prepared by chemically modifying isolated cell membranes using an NHS-PEG-Maleimide linker improved CMCNPs’ diffusion in tumors due to partial decomposition of the tumor matrix [34]. In another study, targeting peptides were exposed by introducing streptavidin to RBC membranes through lipid insertion and incubating with a biotin-linked tumor-targeting peptide. Peptides displayed by the streptavidin–biotin interactions allowed for superior targeting abilities of the modified CMCNPs [35]. Nevertheless, excessive membrane modifications can disrupt the protein and lipid composition of the membranes and lead to immunogenicity of membrane proteins and thus, possible immune reactions against CMCNPs [36].

To form plasma membrane vesicles, which can later be fused with synthetic NPs, the cell membrane pellet is collected and extruded. Cell membranes enclose into vesicles when forced through a porous polycarbonate membrane. RBC membrane-derived vesicles with a hydrodynamic diameter of around 200 nm can be generated by extrusion [37].

Ultrasound waves can be used to produce plasma membrane vesicles as well. Before the sonication process, cells are lysed in an alkaline solution and subjected to pH neutralization. Sonication allows for the self-assembly of cell membrane fragments into membrane vesicles. Sonication has been shown to be an effective method to generate artificial membrane vesicles with high yield [38]. However, sonication may deform cell membranes, cause protein denaturation, and lead to non-uniform vesicle size [39,40].

The produced biomimetic membrane vesicles can also be used for drug delivery per se and have been called “nanoghosts” [41]. Nanoghosts produced from MSCs help in injury and inflammation targeting, which is useful for cancer therapy, while macrophage cell-derived nanoghosts can reduce tumor growth by mimicking macrophage behavior and stimulating the immune response. These nanovesicles demonstrate high potential for drug delivery, combining the simplicity of production, biocompatibility, homogeneity, and high loading capacity [42].

The primary challenges currently faced in this field include the contamination of cell membranes with contents from the cell nucleus and mitochondria, such as proteins and DNA, which can potentially induce harmful inflammatory and autoimmune responses [43]. Notably, double-stranded DNA can be detected in cell membranes at relatively high concentrations when inadequate purification techniques are employed [44]. A critical issue is the selection of the appropriate cell origin for isolating cell membranes, as this choice significantly influences their targeting properties. NPs derived from tumor membranes exhibit specific targeting to particular tumors [45], while membranes obtained from cardiospheres predominantly target the heart [46]. NPs derived from mesenchymal stem cells (MSCs) tend to accumulate in areas of tissue injury and inflammation. Another significant issue is the compatibility of membrane-derived NPs for repeated dosing and universal application in human populations, regardless of HLA alleles. This necessitates the development of a biocompatible cell line, which could be achieved, for example, by deleting the MHC class I loci in either transformed or non-transformed human cell lines [47].

#### 3.1.3. Cell Membrane Coating

The next step is fusing plasma membrane vesicles with NPs, which presumably involves several steps, such as (1) the adsorption of cell membrane vesicles onto the NP surface; (2) the rupture of cell membrane vesicles by mechanical forces and the formation of cell membrane patches; and (3) the fusion of cell membrane patches. Interestingly, cell coating can be oriented right side out or inside out. Such an orientation is not entirely stochastic and depends on the coating technique and experimental conditions [27,48]. The preferred coating is right side out, which preserves the membrane’s surface characteristics. To form a correctly sided membrane coating, the more hydrophobic inner leaflet of cell membranes is directed toward a support surface by the hydrophilic sugar residues of glycoproteins in the outer leaflet [49]. However, an inside-out coating may be useful for screening drugs that can bind to intracellular domains [48].

The physicochemical characteristics of the NP core, such as its size, shape, charge, and elasticity, play a major role in the efficiency of the cell membrane coating and drug loading [24]. NP coating is dependent on the charge of the synthetic NPs and the added components, like the membrane-to-polymer ratio for coating polymeric cores [50]. Strong electrostatic interactions between negatively charged membranes and positively charged NPs induce collapse of the fluidic lipid bilayer and disordered structure, resulting in extensive aggregation, whereas negatively charged NPs can be successfully cloaked with right-side-out membranes [51]. Nevertheless, the spontaneous fusion of negatively charged NPs with cell membrane vesicles is unlikely due to the same charge and presence of proteins in the cell membrane of the latter. Therefore, the preparation of CMCNPs with negatively charged cores often requires the application of external forces (e.g., extrusion) for fusion [52].

Various methods, such as co-extrusion, sonication, microfluidic sonication, microfluidic electroporation, co-incubation, and flash nanocomplexation, can be used to fuse cell membrane vesicles to synthetic NPs. Extrusion is the most frequently used technique; plasma membrane vesicles and NPs are mixed in a certain ratio and added to the extruder. The membrane structure of vesicles is disrupted and reformed around the cores of synthetic NPs. Extrusion offers a consistent size of the synthesized CMCNPs with a relatively high right-side-out orientation ratio [27]. The main disadvantages of extrusion are its significant mechanical stress and high sample waste that leads to the accumulation of material on the extruder’s porous membrane [53].

Sonication is another common method for CMCNP preparation. The application of ultrasound energy ruptures cell membrane vesicles and allows for the spontaneous reassembly of cell membranes around NP cores. In comparison to co-extrusion, sonication results in less material loss and requires less time [10], but results in a lower right-side-out orientation coating ratio, leads to polydisperse coating, and can disrupt the NPs [29,54]. Furthermore, the coating protocol for scalable production is still not standardized [55]. Sonication results in a lower ratio of full coating than extrusion and the combined sonication–extrusion approach [51]. Moreover, neither extrusion nor sonication are applicable for irregularly shaped NPs [56]. To overcome this limitation, PEG-assisted membrane coating is proposed [57].

Microfluidic sonication-based coating was developed to overcome the laborious and time-consuming limitations of co-extrusion and sonication [58]. Microfluidic sonication is a technique in which a microfluidic device is submerged in an ultrasound bath before the CMCNP fabrication process. A notable advantage of microfluidics-based fusion is that the fabrication process of synthetic NPs can be combined with subsequent fusion of the synthetic cores with cell membrane vesicles. CMCNPs with liposome [59] and polymeric [60] cores have previously been synthesized using microfluidic sonication. This technology provides the rapid and controlled assembly of NPs, but the requirement for parallel upscaling makes this approach challenging for large-scale production. Furthermore, it has other disadvantages inherent in conventional sonication.

In microfluidic electroporation, a mixture of plasma membrane vesicles and NPs is added to the microfluidic device, followed by electroporation, which allows for the formation of transient pores in cell membranes and their wrapping around NPs [61]. Microfluidic electroporation is a high-throughput method with a high synthesis rate, and can be scaled up in parallel [61,62]. However, issues such as a lack of standardization, possible batch-to-batch variability, and induced NP aggregation must be addressed [27].

Co-incubation is a non-disruptive method that does not require additional equipment. It is most effective with positively charged NPs and may still result in the aggregation of cell membranes and NPs, non-uniform coating, and uneven CMCNP size [55,63].

To overcome the problem of aggregation between positively charged NPs and negatively charged cell membranes, a strategy called flash nanocomplexation (FNC) was developed for scalable CMCNP manufacturing [55]. FNC utilizes the sonication of the cell membrane vesicles with a subsequent introduction of cell membrane fragments and a NP solution into different inlets of a multi-inlet vortex mixer. Fast and uniform coating is then achieved through dynamic turbulent mixing. CMCNPs prepared from cationic NPs by FNC have better colloidal stability and a smaller size change than CMCNPs generated by bulk sonication. Nevertheless, further research for cell membrane coating of positively and negatively charged NPs is required.

In addition to coating cells with isolated membranes of a single cell type, cell membranes of different origins can be used in CMCNP production. Hybrid CMCNPs can carry properties of both source cells, such as the prolonged circulation time of RBCs and the targeting abilities of tumor cells [64]. Hybrid membranes can be obtained before or after cell membrane extraction. Fusion prior to cell membrane extraction is accomplished by co-culturing tumor cells and another cell line in a PBS solution containing PEG and dimethyl sulfoxide [65]. This creates fused cell lines that possess characteristics of both source cells. The drawbacks of this strategy include the possible formation of unwanted fusion cell lines (A–A or B–B) and the complicated purification of A–B cells, which relies on methods such as fluorescence-activated cell sorting to isolate single cells that exhibit characteristics (e.g., dual fluorescence) present in fused cell lines [66]. Therefore, fusion after cell membrane extraction is the most frequently utilized approach. The most common method for creating hybrid membranes after cell membrane extraction is sonication, where cell membranes from different sources are collected and sonicated to ensure the fusion of disrupted cell membranes.

Incomplete NP coating is a major problem in CMCNP technology, as it results in partial exposure of the NP core to the immune system, reduces biocompatibility, and leads to drug leakage. Moreover, the endocytic entry of CMCNPs is dependent on their coating [51]. More highly coated CMCNPs (≥50% coating) tend to enter cells by themselves, whereas CMCNPs with low coating aggregate before internalization by the cells.

Common characterization techniques can evaluate cell membrane coating efficiency only qualitatively rather than quantitatively. Recently, a fluorescence quenching assay showed that after a standard protocol, over 60% of CMCNPs had less than 20% coating [51].

Apart from the described protocols for CMCNP preparation, another method has been proposed to coat membranes of living cells onto NPs [67]. Silica NPs can be coated with phospholipid bilayers through mechanical penetration into concentrated lipid layers [68]. Later, the same research group coated silica NPs with the cell membranes of living cells [67]. Briefly, cells are concentrated at a specific density layer within the tube by adding a sucrose solution, and then silica NPs are added to the top of the concentrated cell layer. By applying UC, the authors achieve a successful mechanical penetration of silica NPs into living cells, thus coating NPs with cell membranes. The fluorescence quenching assay shows complete coating of 20% of the particles via this method, higher than in the original study, in which standard CMCNP preparation was performed (~10% of NPs fully coated) [67]. This technology has the potential to simplify the relatively complex fabrication process of CMCNPs and requires further research.

### 3.2. Cargo Loading into CMCNPs

NPs are generally pre-loaded with cargo, since loading into formed CMCNPs is usually less efficient and can compromise their structural integrity. The process of loading depends on the type of NP core and the method of its preparation, but most of the time, it is achieved by adding the therapeutic cargo to the reagents before the synthetic cores are formed. For instance, the chemotherapy drug docetaxel was added to polylactic-co-glycolic acid (PLGA) in acetone and precipitated dropwise into water to form docetaxel-loaded PLGA cores [32]. Similarly, in another study, siRNA against PD-L1 and the cancer chemotherapeutic DOX were loaded into PLGA NPs using a different preparation method for polymeric NPs called double emulsion solvent evaporation [69]. In an alternative study, the researchers utilized macrophage cell membrane-coated bovine serum albumin NPs modified with poly-L-lysine, a cationic hydrophilic polymer that allows for the formation of complexes with therapeutic DNA or RNA [70]. Coating with cell membranes hid the positive charge of cationic NPs, protected encapsulated mRNA against RNase degradation, and prolonged NP circulation time.

In addition, sometimes cargo can simply be incubated with synthetic NPs after their formation. For example, magnetic mesoporous silica NPs incubated with the photosensitizing compound hypocrellin B successfully incorporated the drug due to NP porosity [71].

Nanoghost loading is performed as pre-production (during the downsizing of particles) and post-production approaches. In the experiments of Toledano Furman et al. [41], the uploading of soluble TRAIL protein (sTRAIL) was performed by co-extrusion of recombinant protein with cell membranes during nanoghost preparation. Unpacked sTRAIL protein was separated from particles by ultracentrifugation. A Bradford assay demonstrated ~30% sTRAIL encapsulation efficiency. The systemic application of NG-sTRAIL in a xenografted prostate cancer mice model resulted in an 80% tumor growth inhibition [41]. In the experiment by Kaneti et al. [72], the loading of plasmid DNA encoding tumor-inhibiting PEX protein was performed by membrane sonication during downsizing. To reduce plasmid damage by shear stress during sonication, the plasmid was complexed with polyethylenimine. The treatment of cells with loaded nanoghosts reduced cell viability by 30% [72].

The main post-production cargo loading approach is the electroporation of prepared nanoghosts. Oieni et al. [73] performed electroporation for the loading of antisense oligonucleotides (ASOs) into particles followed by size-exclusion chromatography to remove free ASOs. Optimized low-voltage electroporation resulted in a 30% encapsulation efficiency with negligible ASOs or nanoghost aggregation. The application of prepared nanoghosts resulted in a 90% knockdown of target miRNA in MSCs [73]. The developed electroporation protocol was also used for the loading of another type of cargo, including plasmid DNA, small-molecular drugs, and peptides with 73.8%, 26.2%, and 35.3% loading efficiency, respectively [74].

In conclusion, current methods for cargo loading into BMNPs are hindered by nanoparticle damage and low encapsulation efficiency. Therefore, the development of novel and efficient loading techniques based on molecular complexation during the BMNP production process should be prioritized in BMNP research in the coming years.

### 3.3. Artificial Cell-Derived Vesicles (ACDVs)

Artificial cell-derived vesicles (ACDVs) are commonly obtained by extrusion or microfluidics-based cell fragmentation methods, but nitrogen cavitation, cell blebbing, and sonication techniques can be implemented as well. The most frequently used method for ACDV production is the serial extrusion of cells through polycarbonate membrane filters with decreasing pore diameters [75]. Extrusion provides high yield and uniformity in size and can be scaled up, but its disadvantages include membrane deformation, zeta potential changes, the accumulation of material on the porous membranes, and high material loss [62,76]. Some methods have been developed to further improve ACDV yield after extrusion. Subjecting cells to iterative freeze–thaw cycles before the extrusion process can increase ACDV yield by up to three times compared to conventional extrusion [77]. However, this additional freeze–thaw may lead to the denaturation of vesicle proteins. Alternately, using a cell-binding lipid detergent during extrusion increases ACDV yield by approximately 20-fold [78]; this, however, can lead to protein denaturation and changes in membrane integrity and particle content, along with difficulty in detergent removal.

Centrifugation-based extrusion is another method proposed for the large-scale production of ACDVs [79]. The device utilizes centrifugal force and a polycarbonate filter with micro-sized pores to form ACDVs. Centrifugation tends to be less laborious than conventional extrusion and provides 250 times more nanovesicles than natural EV yield. This strategy is further improved by simply using spin cups fitted with membrane filters [13]. The shearing approach using spin cups produces ACDVs with size, morphology, and zeta potential comparable to those of EVs [80].

Microfluidic fabrication has emerged as a simple alternative method for BNP production [81]. Microfluidics-based approaches for ACDV production employ either cell slicing or microfluidics-based extrusion [82]. These approaches are based on the self-assembly of fragments left after the minimization of the free energy of lipid bilayers. This approach makes the ACDV production process more controllable due to precise fluid handling and a reduction in the applied shear stress. The scaling potential of microfluidic fabrication still needs to be determined.

Cells can also be fragmented using nitrogen cavitation. A quick shift in a liquid’s pressure results in the creation of many small cavities filled with vapor, which rupture the cells, and vesicles form through the self-assembly of cell membrane fragments. However, it has the previously mentioned disadvantages, such as its time requirement, the need for expensive equipment, and the possibility of changing the intact structure of the membranes, which limit its use. Furthermore, the increase in yield is not very high (16-fold over the natural EV production rate [83]) compared to other methods such as extrusion (up to 250–500-fold [14,84]) and microfluidics-based approaches (about 100-fold [85]).

A cell bleb-based method was proposed as an alternative to other cell-disrupting methods. In this method, the cells are exposed to a sulfhydryl blocking reagent or cytochalasin D, an actin polymerization inhibitor. While cytochalasin D can directly prevent the addition of further actin monomers, sulfhydryl blocking reagents act by modifying thiol groups of the proteins involved in connecting the actin cytoskeleton to the cell membrane. As a result, the structural integrity of the cell membrane and the cytoskeleton is altered, which promotes formation of blebs (protrusions of the cellular plasma membrane). Initially, the produced vesicles were heterogeneous in size with a predominance of large vesicles, and were thus not appropriate for drug delivery [85]. Optimized sulfhydryl blocking methods using chemical stimulation allow for the production of ACDVs that are homogenous in size [86]. Nevertheless, the difficulty in removing residual reagents and possible damage to cell membranes and their components impede the application of this technique [76]. The relatively low yield increase (about 10-fold over natural EVs) is another drawback of this method [87].

### 3.4. Cargo Loading into ACDVs

Similarly to EV loading strategies, endogenous or exogenous loading can be used for ACDV cargo packaging. Endogenous loading can be achieved by modulating the parental cells. Producing cells can be transduced to overexpress nucleic acids of interest (Figure 2). Transduction is usually performed on cells that are difficult to transfect, such as primary cells [88]. For instance, shRNA against the c-Myc gene was loaded into ACDVs by culturing embryonic mouse fibroblast cells transduced with lentiviruses encoding shRNA [89]. shRNA was then processed within the cell to create siRNA, which was capable of gene silencing and reduced gene expression by approximately 50%.

Passive endogenous loading is another strategy. Cells are pre-incubated with therapeutic cargo before the formation of ACDVs. The desired drug diffuses into the cells because of the concentration gradient across the cell membrane. For example, MSCs derived from human-induced pluripotent stem cells were successfully pre-incubated with docetaxel for 24 h before being processed with serial extrusion [90]. However, passive loading has low efficiency and is mainly used with hydrophobic cargo due to the low ability of hydrophilic drugs to migrate through the lipid bilayer of the cell membrane.

Exogenous loading is an alternative approach performed after ACDV generation. Previously described methods for exogenous EV cargo loading, such as electroporation [89], sonication [91], ACDV transfection [92], chemical permeabilization with saponin [93], and the conjugation of cholesterol to RNA [94], can also be utilized to package molecules exogenously into ACDVs. Additionally, a study investigated the efficiency of DOX loading into ACDVs using four different methods, namely freeze–thaw, saponin permeabilization, incubation at 37 °C for 5 min, and incubation at room temperature for 24 h [95]. The addition of saponin resulted in the highest loading efficiency due to the highest permeabilization of the ACDV membrane. However, higher membrane permeabilization leads to a corresponding increase in vesicle size. Therefore, the balance between maximum drug loading and minimum size increase is desirable.

Passive exogenous loading can be implemented as well. Simply incubating therapeutic cargo with ACDVs leads to its incorporation into vesicles, albeit with a lower efficiency. This method can generally be used for hydrophobic cargo due to the low permeability of the ACDV membrane for hydrophilic cargo. For instance, hydrophobic DOX can be loaded into ACDVs via shaking for 2 or 12 h at 37 °C, depending on the protocol [86,96].

## 4. Bottom-Up Approach

In this section, two biomimetic bottom-up techniques will be discussed as examples, focusing on the membrane functionalization of liposomes and fully synthetic vesicles.

### 4.1. Liposome Modification with Cell Membrane Proteins

One way to endow NPs with additional properties (e.g., specific targeting and evasion of macrophage clearance) is to conjugate a specific protein to the surface of synthetic NPs. Liposome surface modification with cell-penetrating peptides or antibodies can facilitate the delivery of therapeutic agents by enhancing endosomal escape or targeting, respectively [97,98]. On the other hand, conjugation with pH-responsive peptides can provide trigger-responsive properties [99].

A more sophisticated strategy relies on incorporating multiple cell membrane proteins into the liposome bilayer. Cell membrane proteins define the surface interactions of the cells, helping the cell to interact with its environment, contributing to its structural integrity, and lowering immunogenicity. Incorporating cell membrane proteins into liposomal surfaces was first performed with integrated proteins derived from the leukocyte plasmalemma [100]. The resulting liposomes containing integrated proteins were named leukosomes, and they were designed to mimic natural cells by exploiting intrinsic properties of leukocyte membrane proteins. Since the development of leukosomes, proteins from other sources, like cancer cells [101] and RBCs [102], have been incorporated into liposomes. For instance, hybrid membrane proteins from different cell sources incorporated into synthetic phospholipid bilayers combine the ability of RBC membrane proteins to protect NPs from phagocytosis with the targeting ability of cancer cell membrane proteins [102].

Biomimetic liposomes produced by the bottom-up approach are composed of cell membrane proteins and liposomes. Cell membrane proteins are isolated before liposome fabrication. Currently available extraction kits, such as ProteoExtract^®^ (Merck Millipore, Burlington, VT, USA) and Mem-PER™ (Thermo Fisher Scientific, Rockford, IL, USA) Plus, are the most common approach for obtaining cell membrane proteins. The main concept is to permeabilize the cell membrane by adding the lysing solution, remove cytosolic proteins after centrifugation, and solubilize the membrane proteins in a buffer, which then can be collected after the centrifugation step. Membrane protein extracts can be stored at −20 °C. The major disadvantage is that commercial kits are rather expensive to use. Therefore, an optimized detergent-based protocol for the extraction of cell membrane proteins can be used as an alternative [103], with yields and purity similar to those obtained with commercial kits.

The second component are the liposomes, which are synthetic spherical NPs formed using amphipathic phospholipids. Liposomes have an aqueous core and a lipophilic shell, which allow them to encapsulate both hydrophilic and hydrophobic drugs. Other molecules, such as steroids, surfactants, charged lipids, and polymers, can be utilized to endow liposomes with different properties [104]. For instance, cholesterol changes membrane fluidity and improves its stability, whereas PEG is often used to prolong liposome circulation half-life [105]. Liposomes are the most explored and widely used drug delivery system, and numerous liposomal formulations for treating diseases have received FDA approval.

Methods for preparing liposomes have already been covered in other comprehensive reviews [106,107]. This section will focus on the methods that have been utilized to prepare liposomes for subsequent integration of cell membrane proteins, such as thin-film hydration and microfluidic fabrication.

Thin-film hydration, also known as the Bangham method, is the most widely used technique for liposome preparation. Lipids are mixed with an organic solvent (usually chloroform), which is then evaporated to form a dried lipid film. The film is rehydrated with an aqueous solution (e.g., PBS) containing cell membrane proteins. The thin-film hydration technique is simple, straightforward, and does not require expensive equipment, but the major problem of this approach is the difficulty in completely removing the organic solvent, which can influence the stability of the generated liposomes and perturb chemical properties of the incorporated molecules [107]. Furthermore, this technique is time-consuming, difficult to scale up, creates large liposomes with broad size distribution, provides low entrapment efficiency of the hydrophilic drugs, and requires sterilization for clinical use [108,109].

Conventional thin-film hydration results in the production of relatively large vesicles, including giant (>1 μm) or large (200 nm–1 μm) unilamellar vesicles (GUVs and LUVs, respectively) [108]. Additionally, multilamellar vesicles (MLVs), comprising multiple phospholipid bilayers, are obtained. Methods to break down large vesicles and MLVs and form small unilamellar vesicles (SUVs) in the desired size range have thus been devised. Extrusion is widely used for this purpose, but techniques like sonication, freeze–thaw (alone or in combination with sonication), and homogenization can be utilized as well [108]. Compared to other methods, extrusion shows great results in size reduction, producing homogenous liposomes [110]. Furthermore, an alternative method to integrate cell membrane proteins into liposomes is after hydration of the thin film during the extrusion process. Cell membrane proteins can be embedded in the lipid bilayer of liposomes during the extrusion of already formed MLVs in order to obtain SUVs with integrated membrane proteins [111]. The main disadvantages of extrusion are the high material loss and potential clogging of the membrane filter, which can be reduced by a higher extrusion flow rate and a more diluted liposome suspension [107,110].

The microfluidics-based formation of liposomes is an alternative approach to conventional thin-film hydration. NanoAssemblr™ and NanoGenerator™ Flex-M platforms have already been used to generate liposomes with incorporated cell membrane proteins [112,113]. These platforms allow microfluidic mixing in herringbone channels. In the device, the cartridge is connected to two inlet streams. One contains phospholipids dissolved in a water-miscible organic solvent, and the second contains an aqueous solution with suspended cell membrane proteins. Controlled mixing of the organic and aqueous solutions induces the self-assembly of the hydrophobic lipids into liposomes [114]. Shear stress caused by the microfluidics components breaks down the generated vesicles into liposomes with a narrow size distribution. Around 90% of cell membrane proteins added to the aqueous stream associate with the final formulation compared to 63% using the thin-film hydration method [112]. The microfluidics-based approach allows for the rapid, controlled, and homogenous mixing of components under gentle conditions with no requirement for further downsizing. With this strategy, continuous but small/medium scale production can be achieved by upscaling in parallel [115]. One possible drawback is groove blockage by particles, leading to sample flow stagnation. Furthermore, microfluidic conditions, including the total flow rate, flow rate ratio, and lipid ratio/concentration, should be optimized for each system [116].

### 4.2. Cargo Loading into Liposomes with Incorporated Cell Membrane Proteins

Cargo can be loaded into biomimetic liposomes with incorporated membrane proteins passively or by using remote loading. Hydrophilic therapeutics can be added to the aqueous solution along with cell membrane proteins, whereas hydrophobic drugs can be added to the organic phase. For example, the immunosuppressive drug rapamycin is added to the organic phase before the synthesis of liposomes by a microfluidic approach [117]. In another study, artificial long intergenic non-coding RNA (lincRNA), which regulates inflammation and promotes neuron regeneration, can be successfully loaded into leukosomes by hydrating the lipid film with a solution of proteins and lincRNA [118].

Another approach is the remote loading of therapeutics. In addition to the pH gradient modification described above as a strategy for EV loading, another remote loading approach based on the transmembrane gradient of ammonium sulfate can be utilized. When liposomes are generated by hydration/microfluidic mixing in an ammonium sulfate solution, this salt is present in both the intra- and extraliposomal aqueous phases [119]. An ammonium sulfate gradient creates a transmembrane pH gradient and acts as the driving force for the efficient and stable remote loading of amphipathic weak bases (e.g., DOX) into liposomes [120]. Inside the liposomes, therapeutic molecules become positively charged, losing the ability to permeate the liposomal membrane. Two studies used the remote loading of DOX into leukosomes [114,121]. Before DOX loading, the aqueous phase of the formulation, consisting of 250 mM of ammonium sulfate buffer with diluted cell membrane proteins, is mixed with the organic phase using the microfluidics-based system Nanoassemblr™. The DOX solution is then simply incubated with the already formed leukosomes at 37 °C for about 1–2 h, allowing for the incorporation of the drug into the lumen of the particles.

### 4.3. Fully Synthetic Vesicles

Lipid composition also plays a major role in vesicle functionality. Along with proteins, lipids are important membrane constituents, providing the proper microenvironment for the attachment of membrane proteins and contributing to membrane tension, rigidity, and carrier overall shape [122]. Alterations in lipid composition can dramatically affect the interaction of nanoparticles with the immune system. For example, redistributing phosphatidylserine to the outer leaflet of the plasma membrane can result in the attenuation of the immune system response [123]. Coating NPs with lipid bilayers also changes their intrinsic properties; coating inorganic NPs with lipids can improve their stability and reduce adverse effects in vivo [124].

It is worth noting that, similar to cell membranes, EV membranes are composed of several different kinds of lipids. However, the main lipids commonly found in EVs are cholesterol, sphingomyelin, phosphatidylcholine, phosphatidylserine, and phosphatidylethanolamine [125]. By using a combination of different types of lipids, EV-mimicking lipid-based NPs can be created to imitate naturally derived EVs. The obtained BNPs are more homologous and less variable than naturally derived EVs. Several EV mimetic formulations have been created over the past few years [126,127,128,129], demonstrating reduced cytotoxicity, increased storage stability, and cellular internalization efficiency after the proper adjustment of liposome lipid composition. Still, these formulations do not present the optimal EV lipid composition.

Fully synthetic vesicles (fSVs) offer several advantages over natural EVs, including precise control over their composition and size. Similar to biomimetic liposomes (e.g., leukosomes), liposomes (or more precisely, SUVs) form the basis of fSVs. To prepare fSVs, one group used a composition closest to the natural EV lipid composition [130]. However, to achieve vesicle sizes larger than SUVs, the study added shear stress emulsification, a method that has shown promising results for preparing droplet-stabilized GUVs for the assembly of cell-like compartments [131]. The researchers implemented several changes from the GUV protocol to generate smaller vesicles (median diameter of 400–600 nm), including increasing the speed and duration of the emulsification procedure and increasing concentrations of MgCl_2_, negatively charged surfactants, and lipids [132].

One of the main steps of this production protocol is the emulsification of the oil and water phases to obtain a water-in-oil emulsion with incorporated SUVs, which can later be fused. SUVs are prepared by conventional thin-film hydration followed by extrusion, while the oil phase is composed of neutral and negatively charged surfactants and FC-40 oil, which has a high viscosity [133]. The aqueous phase is added on top of the oil phase and the whole mixture is then emulsified using an emulsifier at high speed. Emulsification leads to the formation of surfactant-stabilized water-in-oil droplets, at the periphery of which SUVs undergo charge-mediated fusion, creating larger vesicles with a median diameter below 1 μm [134]. To release vesicles from the oil phase and the surfactant shell, PBS enriched with the de-emulsifying surfactant is added to the emulsion [132].

Additionally, to better imitate EVs, vesicles can be functionalized by adding miRIDIAN miRNA mimics to the aqueous phase prior to emulsification to imitate endogenous miRNAs [130]. The synthetic lipid composition also includes NTA (Ni2+)-modified lipids, which specifically interact with histidine-tagged CD9, CD63, and CD81 proteins. Due to the histidine–nickel interaction, added tetraspanins are incorporated into the bilayer of obtained vesicles in the correct orientation, ensuring better compatibility and less immunogenicity of fSVs.

Despite the enthusiasm surrounding fully synthetic vesicles (fSVs), their composition remains relatively simple, with membrane contents resembling those of liposomes. The complexity of natural EVs, ACDVs, and CMCNPs underlies their ability to effectively traverse biological barriers and be internalized by various cell types. The intricate array of surface ligands and receptors present in these natural systems is not replicated in fSVs; thus, fSVs serve primarily as surrogates for BMNPs, with applications limited to specific contexts. Nevertheless, fSVs represent a significant advancement compared to organic nanoparticles, and their production process closely resembles that of synthetic nanoparticles.

### 4.4. Cargo Loading into fSVs

Hydrophilic therapeutics can be added to the aqueous phase before layering it on top of the oil phase in the same manner as the encapsulation of synthetic miRNAs [130]. However, no study has yet loaded hydrophobic molecules into fSVs. The addition of hydrophobic molecules to the oil phase may disturb droplet stability and vesicle formation and more testing is needed. An alternative way to incorporate hydrophobic molecules can be simply by co-incubation after fSV production.

## 5. Purification of BMNPs

BMNP production results in undesirable contaminants that may affect BMNP quality for downstream applications. This is especially important for ACDVs because the conventional extrusion method for their preparation frequently leads to high levels of impurities. In contrast, nanoghost and CMCNP production result in the elimination of most parts of the contaminants during membrane isolation. Leukosomes and fSVs requires the elimination of residual solvents and lipids, and leukosomes require the discarding of unincorporated membrane proteins. Additionally, the loading of desired cargo in BMNPs also requires the discarding of unpackaged molecules and even particles (in the case of CMCNPs). Additionally, purification methods can improve the homogeneity of obtained BMNPs. For the purification of BMNPs, some approaches developed for extracellular vesicles can be used due to evident similarity in particle composition (Figure 3).

Differential centrifugation is a “gold standard” in extracellular vesicle isolation. The method is based on the use of centrifugal forces to pellet vesicles, while other contaminants remain in the supernatant. Differential centrifugation includes the following three steps: centrifugation at 500× *g* (pelleting cells), 2000× *g* (pelleting cell debris), 10,000× *g* (pelleting large vesicles), and 100,000× *g* (pelleting EVs). This has evident disadvantages, including vesicle damage, low yield, bad purity, and highly limited scalability [135,136,137,138,139,140,141]. Due to the inefficient elimination of impurities, it can be applied for the purification of BNPs with a low number of technological contaminants, for example, nanoghosts. Also, this method can be used for discarding some types of unpacked cargo (small molecules, therapeutic RNAs, small proteins, and others).

Density gradient ultracentrifugation is based on the application of a media gradient, resulting in retaining particles in media fraction with an appropriate density during the centrifugation process. The density gradient ultracentrifugation method is non-damaging for particles and effectively removes different types of contaminants, resulting in very high purity [14,135,142]. For this reason, this approach is the most commonly used for ACDV purification. The main disadvantages of this method are its low scalability and the requirement of an additional purification step to remove media remnants from the isolated sample.

During ultrafiltration, a sample is passed through a semi-permeable membrane under pressure or by centrifugation. The membrane cut-off exceeds the size of particles (usually less than 750 kDa), resulting in the removal of impurities into the filtrate, while nanoparticles are collected from the retentate fraction. Ultrafiltration is widely used for industrial-scale purification and concentrating proteins and thus is easily scalable. The main disadvantage of ultrafiltration is possible “cake formation”, resulting in the accumulation of large particles at the membrane surface leading to the diminished filtration of contaminants [14,135,143,144,145,146,147].

In the tangential flow filtration method, fluid stream is directed tangentially to the semi-permeable membrane, and the additional flow gradient is directed toward the membrane. Small contaminants are filtrated through the semi-permeable membrane, while large particles are retained in the filtration system and collected. “Cake formation” is reduced in TFF, as large molecules are washed off from the filter surface by tangential flow. TFF is also widely used for industrial-scale protein purification and can be easily adopted for nanoparticles [14,148,149,150,151,152].

Size exclusion chromatography (SEC, or gel filtration) is based on separating molecules and particles in a column filled with porous sorbent. The size of large particles exceeds the diameter of the pores and thus, large particles are not entrapped in sorbent beads and eluted first, whereas small particles or molecules are reversibly absorbed inside of the beads and eluted later. As the optimized SEC procedure eliminates up to 99% of non-vesicular proteins, this method can be used for highly contaminated samples. The disadvantages of SEC are sample dilution and the requirement of pre-concentration [153,154,155,156,157]. Bind–elute chromatography is a modified SEC that uses porous sorbents with an absorptive core. Molecules smaller than pores enter the beads and tightly bind the absorptive core, resulting in strong entrapment, whereas particles large than pore diameters pass through the column. Bind–elute chromatography results in better purity than SEC and a less diluted sample [158].

The optimal approach for BMNP purification is determined by the type and amounts of contaminants. BMNP purification has similarities to EV isolation methods, and techniques like UF [92,96], TFF [77], SEC [93], and centrifugation [159,160] can be used. The optimization of the process is often needed due to differences in BMNPs’ composition, density, and surface properties. Differential ultracentrifugation results in low purity and cannot be scaled; thus, its application for BMNP purification is limited. DGU is a main approach for the purification of ACDVs in published papers, and this method can also be used after ACDV production to enhance size homogeneity and sample purity of the obtained vesicles [81]. Nevertheless, this approach is also unscalable and requires the additional removal of residual toxic gradient media and thus is incompatible with good manufacturing practice. UF and TFF techniques are favored for their simplicity and capacity to process large volumes. Moreover, these methods can concentrate the sample. However, since these methods can still lead to contamination, integration with other purification techniques like SEC would be beneficial to obtain purer BMNP fractions. SEC results in high-purity BMNPs and preserves their integrity; thus, SEC and bind–elute chromatography are optimal methods for the purification of all types of BMNPs after combining with UF/TFF as a pre-concentration step.

## 6. Technological Challenges of BMNPs

### 6.1. CMCNPs

For CMCNPs to maintain their biomimetic properties, the integrity of the coating membranes must be preserved. Alterations in the membrane’s integrity impact the biological functions of CMCNPs. Currently, keeping the membranes intact is difficult because existing isolation methods require harsh conditions. Therefore, new simple methods with a negligible impact on the integrity of the membranes are needed.

As mentioned, low coating efficiency can be a challenge for CMCNP applications. Partial coating is associated with limited fluidity of the cell membranes, which leads to failure of the fusion of adjacent cell membrane patches [52]. To increase the degree of cell membrane coating, hybrid membrane vesicles can be produced by co-extruding cell membrane vesicles and liposomes containing a helper lipid (DOPC). Then, hybrid membrane vesicles can be co-extruded with synthetic NPs to form CMCNPs with the desired core. The introduction of a helper lipid increases the membrane fluidity of the original cell membrane vesicles, improving cell membrane coating, but higher membrane fluidity can affect CMCNP functionality (e.g., membrane stability).

### 6.2. ACDVs

One disadvantage of ACDVs is contamination with cellular material during their manufacturing process. Because ACDVs are generated by disintegrating whole cells, intracellular components are encapsulated within ACDVs. Stochastic vesicle formation makes it difficult to predict which molecules are thus incorporated.

Since ACDVs are artificially created vesicles, they have no defined markers. ACDVs are generated from diverse lipid sources like plasma and subcellular organelle membranes and have marker signatures distinct from EVs [161]. ACDVs are enriched with proteins localized in endo-lysosomal membranes (LAMP1, LAMP2, SCARB2), membranes of the endoplasmic reticulum (calnexin), Golgi apparatus membranes (GM130), and the plasma membrane (integrin beta-1, flotillin-1, nicastrin) [161,162]. Notably, ACDVs contain more CD63 than EVs, likely because CD63 associates with membranes of intracellular vesicles in MVBs, lysosomes, and at the plasma membrane, besides enrichment within EV membranes [163]. Nevertheless, as each cell type may present distinct markers, the issue of undefined markers for ACDVs still remains.

Another issue of ACDVs that is also common to EVs is the low encapsulation (generally less than 30%) of therapeutics by endogenous or exogenous loading strategies, especially of hydrophilic and negatively charged molecules [81]. Therefore, new loading techniques with enhanced encapsulation efficiency or the identification of additional compounds to improve drug incorporation are needed.

### 6.3. Bottom-Up Strategies

A major problem in fSV synthesis is the inability to replicate the complexity of the EV membrane. EV composition can include hundreds of different proteins and lipids, whereas current strategies for SV production involve only a small number of lipid and protein types [164]. Currently, many questions remain unanswered regarding the significance of different EV components, including which components are responsible for cargo transfer, high tropism, and the low immunogenicity of EVs. It is essential to identify and utilize the fundamental molecules in order to imitate EVs as much as possible without oversimplifying or complicating the system.

There is also the risk of changing the orientation of membrane proteins during their incorporation. Incorrectly integrated membrane proteins can lead to the loss of the desired properties of BNPs and may elicit immunogenicity [81].

### 6.4. Common Challenges

Current strategies for BMNP production remain expensive and most require specialized equipment. Table 1 provides a comparison for the technological aspects of EV and BMNP manufacturing processes. Except for fSVs, manufacturing BMNPs requires cells. Many cells are needed to obtain ACDVs as well as cell membranes and cell membrane proteins for CMCNPs and liposomes with integrated membrane proteins, respectively. A blood supply needed for obtaining cell membranes or isolated proteins from blood cells may also be costly [54]. The BMNP manufacturing process must be streamlined to minimize costs.

The loading of therapeutic cargo into BMNPs has several limitations with each type of BMNP due to differences in particle production protocols and structure. During CMCNP preparation, different types of nanoparticles are used for coating by membrane. Conceptually, particles of any size can be used for CMCNPs. At the same time, it has been demonstrated that the clearance of vesicles from the bloodstream is size-dependent, with particles larger than 200 nm being quickly sequestered by macrophages [165]. Thus, the recommended size of the delivery vehicle should not exceed 150 nm. Given that the thickness of the lipid bilayer is ~10 nm, the inner diameter of the luminal space does not exceed ~120–130 nm [166]. Consequently, the membrane coating of nanoparticles larger than 100 nm may negatively impact the pharmacokinetics of CMCNPs.

Almost all types of BMNPs can be utilized for the delivery of cargo DNA, particularly short RNA. In the case of CMCNPs, therapeutic nucleic acids can be adsorbed onto positively charged nanoparticles prior to membrane coating. For artificial ACDVs, nucleic acids can be incorporated using endogenous methods. In biomimetic liposomes and fully synthetic vesicle technologies, packaging can be achieved through the thin-film hydration of lipids in the presence of nucleic acid solutions [118,130]. It should be noted that the use of sonication during thin-film hydration can damage large molecules, such as plasmids. Consequently, this method may be less feasible in such cases or may require additional pre-complexation with polymers, such as PEI. Additionally, optimized electroporation protocols can be applied to nearly all types of BMNPs, including ACDVs, nanoghosts, biomimetic liposomes, and fully synthetic vesicles. However, it is important to consider the potential for vesicle aggregation and cargo damage during electroporation.

Therapeutic proteins represent another significant cargo for delivery via biomimetic systems. Nanoghosts are specifically optimized for protein delivery, and efficient packaging can be achieved during the production of vesicles [41]. ACDVs and nanoghosts can be effectively loaded with proteins using electroporation; however, potential vesicle damage must be considered [41]. The packaging of proteins using bottom-up approaches presents technical challenges.

Another important cargo for targeted delivery is small molecules. The coating of nanoparticles loaded with hydrophilic and hydrophobic molecules can be utilized for small molecule delivery via CMCNPs [32]. The effective loading of small molecules into nanoghosts and ACDVs can be achieved through electroporation and other exogenous physical and chemical methods, including sonication, freeze–thaw cycles, and saponin permeabilization [74]. Hydrophobic drugs are incorporated into nanoghosts and ACDVs by dissolving them in the membrane; however, this process requires aggressive organic solvents that can damage the vesicle membranes. Consequently, most packaging protocols rely on the use of the salt form of nominally hydrophobic drugs in aqueous solutions (for example, doxorubicin hydrochloride). In contrast, the hydration of lipid films in bottom-up approaches is much more effective for packaging hydrophobic drugs, as the cargo compounds can be mixed with lipids during the lipid film preparation [117]. The relative feasibility of BMNP packaging for different types of cargo is summarized in Table 2.

The complexity of the BMNP manufacturing process and the lack of large-scale preparation methods are current significant problems that impede the clinical translation of BMNPs (Figure 4). Current strategies for membrane extraction are tedious and difficult to scale up. Protocols for the bottom-up assembly of BMNPs are also laborious and require further optimization. ACDVs are more promising because their production requires fewer steps and less time than EV preparation. Nevertheless, BMNPs must still be purified after assembly, reducing final yield and complicating the manufacturing process. Despite these challenges, ongoing advancements hold promise for simplifying these processes.

Additionally, BMNP storage conditions must be optimized. The prolonged circulation time and low immunogenicity of BMNPs depend on the characteristics of the cell membrane and membrane proteins, which can be altered during long-term storage [30]. Therefore, determining the optimal storage conditions of BMNPs to preserve the integrity of the cell membrane and its intrinsic properties is crucial. Determining optimized buffer systems that can stabilize BMNPs during storage and the possible application of lyophilization techniques should be considered to enhance the shelf life of BMNPs.

Cell characteristics can change at different growth stages, altering characteristics of cell membranes and cell membrane proteins and leading to batch-to-batch variability [30]. The poor control of homogeneity of the final BMNPs is another issue that needs to be solved for clinical translation. To address these challenges, the use of cells derived from reliable and safe sources, such as immortalized induced pluripotent stem cells or MSCs, can provide more consistent BMNP quality. From a regulatory perspective, adequate quality control and characterization of BNPs to monitor their physicochemical properties and batch-to-batch variability is necessary [62].

The safety and long-term toxicity of BMNPs must be thoroughly evaluated. BMNPs can trigger an immune reaction, especially if they are produced using cell membrane proteins or cancer cell membranes. When obtaining cancer cell membranes, the nucleus and genetic material should be stringently removed to eliminate potential toxicity and carcinogenicity [30]. Additionally, employing stable cell lines from which the major histocompatibility complex and other heterogeneous antigens have been removed can further reduce immunogenicity and enhance biocompatibility [167]. It is also important to screen RBCs for blood group compatibility before using them for BMNP manufacturing, as RBC hemolysis and activation of the host immune system can be caused by blood group mismatch [54].

The behavior of different BMNPs at the cellular level is still not fully understood. More research is required to fully understand their biological characteristics related to blood circulation, immune response, biodistribution, and elimination [168].

The use of BMNPs is still in its infancy, as they have not been applied clinically. Clinicals trials of BMNPs are more difficult than those of traditional small molecular drugs [169]. It seems reasonable to posit that the transition to clinical trials for BMNPs will encounter challenges similar to those faced with EVs [170,171]. These challenges include complex manufacturing and standardization processes that comply with GMP, a lack of large-scale production capabilities, insufficient yield, and difficulties in preservation [172]. A significant advantage of BMNPs is their potential for large-scale production, which is not feasible for natural EVs. Technological advancements, such as extrusion and microfluidics-based methods, facilitate the industrial-scale manufacturing of BMNPs [27]. The standardization of BMNPs is superior to that of EVs, which often exhibit variability in size, composition, and content due to biological factors. The controlled production processes for BMNPs ensure uniformity in size, shape, and surface properties, which is essential for reproducibility in clinical applications.

The incorporation of functional elements, such as ligands and targeting proteins, enhances the ability of BMNPs to interact with specific cells or tissues. For example, using membranes derived from macrophages or red blood cells can improve targeting capabilities while evading immune clearance. In contrast to the complex and labor-intensive processes required for natural EV separation, the production of BMNPs offers a more cost-effective and time-efficient approach [10].

However, one potential risk associated with BMNPs is the possibility of contamination with cellular contents that may trigger immune responses [43]. The incorporation of cellular debris or unwanted proteins from the donor cell line into BMNPs could induce inflammation upon introduction into the body. Therefore, it is essential to implement rigorous purification and quality control processes to mitigate this risk. The likelihood of an immune response being elicited is contingent upon the source of the cell membranes used for coating. This risk is particularly pronounced when the membranes contain MHC molecules or other immunogenic components [167].

Additionally, the processes employed to prepare BMNPs, including sonication and homogenization, may damage the therapeutic cargo, potentially reducing its efficacy [118,130]. The incomplete coating of nanoparticles with cell membranes may occur, leading to diminished functionality and efficiency in targeted delivery. Such incomplete coverage could compromise the biocompatibility and targeting capabilities of BMNPs [51].

## 7. Pharmacokinetics of BMNPs

One of the most important advantages of natural EVs and biomimetics over synthetic liposomes is the increased bioavailability that results in a prolonged vesicle circulation time and decreased macrophagic clearance.

Several studies describe biodistribution and circulating levels of EVs in detail [173], but information about kinetics and the half-life of ACDVs and BNPs obtained by the bottom-up approach is still insufficient. At the same time, several experiments have been performed to investigate the half-life of CMCNPs. The results of key studies on the pharmacokinetics of NPs and membrane-coated NPs are summarized in Table 3.

When natural EVs are administered intravenously, they are rapidly cleared by MPS, exhibiting a biphasic curve with distinct distribution (α) and elimination (β) phases [173]. The biphasic elimination profile can be mathematically described by a model in which the body is divided into central and peripheral compartments. First, the particles are distributed throughout the central vascular compartment, i.e., through highly perfused organs, such as the liver, kidneys, and lungs. The particles are then distributed to tissues in which drug distribution occurs more slowly (peripheral compartment). The distribution (t_½_α) and elimination (t_½_β) half-lives, respectively, can be used to describe biodistribution in central and peripheral compartments.

CMCNPs typically show a slow initial phase (α), causing the plasma concentration–time curve to become more monophasic. But often, CMCNPs have a relatively fast distribution phase, possibly due to the clearance of CMCNPs with coating defects or with inside-out membrane coating [34,52], and are thus often described using a two-compartment model. We have provided several examples of studies in which the circulation half-life of CMCNPs is described by a two-compartment model (Table 3).

CMCNPs show a longer elimination half-life (from 1.6-fold to 9.5-fold), a higher area under the curve (AUC), higher mean residence time (MRT), and lower clearance compared to non-camouflaged nanoparticles, leading to a general trend in prolonged plasma half-life. Additionally, hybrid membrane-coated NPs have a comparable circulation half-life as CMCNPs coated with membranes of one cell type [64].

ACDVs produced by chemically induced blebbing have an intermediate circulation time (t_½_ = 41 min) [86]; in this study, circulation half-life was calculated without any sophisticated model.

The circulation half-life of liposomes with integrated cell membrane proteins has also been reported. In one such study, the pharmacokinetic parameters of liposomes with incorporated cancer cell membrane proteins, RBC membrane proteins, and hybrid membrane proteins were evaluated using a two-compartment model [102]. Biomimetic formulations significantly extended the elimination half-life (t₁/₂β) compared to bare liposomes. Liposomes integrated with cancer cell membrane proteins had a t₁/₂β of 30.51 h, while those with RBC membrane proteins and hybrid proteins had a t₁/₂β of 17.71 h and 1.55 h, respectively. All these values were considerably higher than the 0.19 h measured for bare liposomes. Interestingly, biomimetic liposomes exhibited short distribution half-lives (t₁/₂α) in all cases, ranging from 0.16 h to 0.39 h.

In conclusion, these findings demonstrate that biomimetic delivery systems offer clear advantages over synthetic carriers in terms of bioavailability. Membrane proteins appear to play a crucial role in this process.

**Table 3 pharmaceutics-16-01306-t003:** Studies utilizing a two-compartment model in CMCNP research.

	Elimination Half-Life (t_½_β; h)	Area under the Curve (AUC; mg·h/L)	Mean Residence Time (MRT0^−∞^; h)	Total Body Clearance (Cl, L/[h/kg])	Reference
RBC membrane-coated polymeric NPs vs. PEG-coated polymeric NPs	39.6 vs. 15.8	NC	NC	NC	[174]
RBC membrane-coated polymeric NPs vs. polymeric NPs	32.8 vs. 5.6	122.3 vs. 16.3	45.8 vs. 6.1	0.04 vs. 0.45	[175]
Cell membrane-coated SiO_2_ NPs vs. SiO_2_ NPs	18.5 vs. 9.1	1345 vs. 360	22.7 vs. 12.2	0.005 vs. 0.017	[52]
PLT membrane-coated nanogel vs. nanogel	32.6 vs. 5.6	112.8 vs. 3	NC	NC	[176]
RBC membrane-coated gold nanocages vs. polymer-coated gold nanocages	9.5 vs. 1	NC	NC	NC	[177]
RBC–PLT hybrid membrane-coated NPs vs. RBC membrane-coated NPs vs. PLT membrane-coated NPs	51.8 vs. 42.4 vs. 38.3	NC	NC	NC	[64]
Cancer cell membrane-coated mesoporous organosilica NPs vs. mesoporous organosilica NPs	18.4 vs. 6.6	NC	NC	NC	[178]
RBC membrane-coated polymeric NPs vs. polymeric NPs	10.7 vs. 6.6	NC	NC	NC	[179]

NC: not calculated.

## 8. Conclusions and Perspectives

EVs play a crucial role in cell-to-cell communication by transporting nucleic acids, proteins, and various types of cargo. Since their discovery, EVs have been recognized as promising delivery carriers capable of encapsulating diverse therapeutic agents. Numerous clinical trials are currently underway to explore a wide range of therapeutic applications. However, several limitations hinder the clinical use of EVs, including low yields, high heterogeneity, and challenges in standardization. These factors complicate the approval process for medical applications involving EVs.

To address these issues, the production of BMNPs has emerged as a viable solution, combining the high bioavailability of natural vesicles with more efficient and cost-effective manufacturing processes. Various manufacturing approaches for BMNPs have been developed, including CMCNPs, nanoghosts, artificial cell-derived vesicles, also referred to as ACDVs, and liposomes integrated with membrane proteins. Establishing standardized cell sources for optimal biomimetic production is essential for advancing BMNP technologies. Consistent and efficient cell lines are necessary to ensure uniform BMNP production.

These standardized cell lines can be further modified to incorporate specific surface elements, such as targeting proteins, “don’t eat me” signals, or ligands for surface receptors, particularly those that require a membrane-bound state for functionality. Additionally, developing cell lines that lack MHC-I molecules can help mitigate immune responses, enhancing the safety of BMNPs for therapeutic use.

Another significant advantage of BMNPs is their ability to encapsulate content that is either inaccessible or ineffective when delivered via exosomes. BMNPs can accommodate a wide range of cargo, including nanoparticles, viruses, and gene-editing tools such as CRISPR and RNA interference (RNAi). BMNPs exhibit distribution kinetics comparable to those of natural EVs and often outperform liposomes in various contexts.

In terms of production, BMNPs offer substantial advantages over exosomes in cost, time efficiency, and yield. The production processes for BMNPs are faster, more scalable, and less expensive, making them more suitable for large-scale therapeutic applications. Furthermore, BMNP production can be precisely controlled, resulting in nanoparticles with consistent size and composition—critical factors for therapeutic efficacy and regulatory approval.

Enhancements in purification and isolation methods are necessary to improve the scalability and purity of BMNPs. Advanced techniques such as chromatography, tangential flow filtration, and affinity-based methods could effectively remove unwanted biological components without compromising the integrity of the nanoparticles.

Moving forward, the development of BMNPs must focus on optimizing production processes, improving cargo loading efficiency, and enhancing vesicle surface functionalization. Addressing these challenges is essential for creating next-generation drug delivery platforms.

## Figures and Tables

**Figure 1 pharmaceutics-16-01306-f001:**
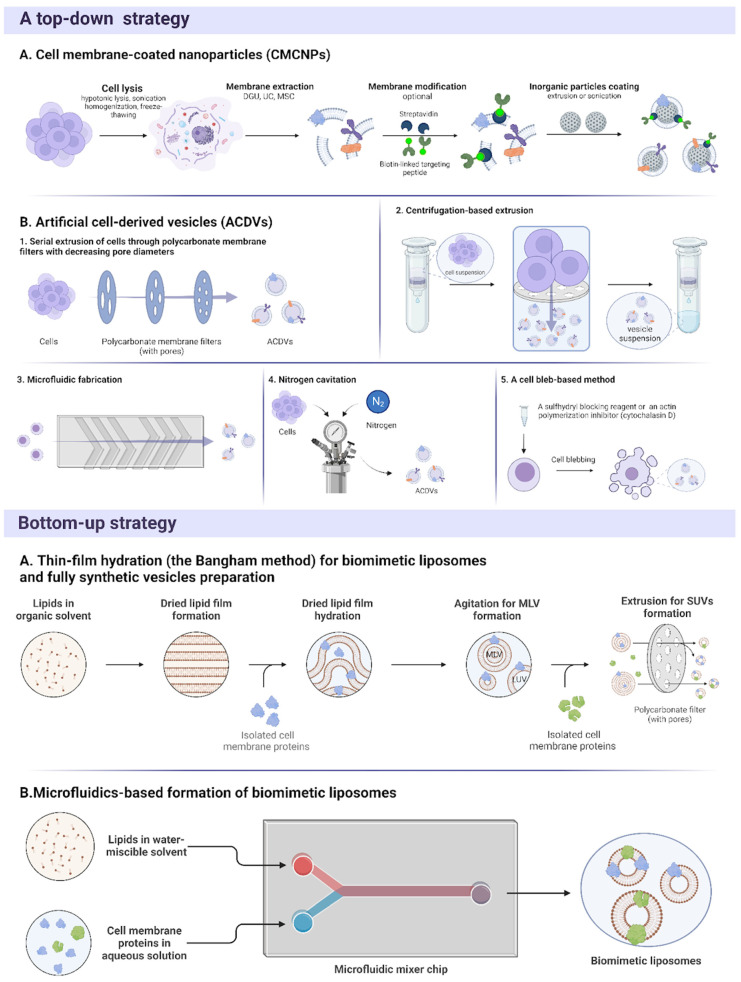
Top-down and bottom-up strategies for BMNP preparation. Top-down strategy: (**A**) preparation of cell membrane-coated nanoparticles, where cells undergo cell lysis by sonication, freeze–thaw, or hypotonic lysis. After the lysis, the membranes are extracted by density gradient ultracentrifugation (DGU), ultracentrifugation (UC), or medium-speed centrifugation. Optionally, the membrane can be decorated with streptavidin beads and biotin-fused peptides can be conjugated onto the membrane. Next, the core particle is coated with a membrane by co-extrusion or sonication. (**B**) 1. ACDVs can be produced by the manual extrusion of cells through membrane filters with decreasing pore sizes. 2. For large-scale production, the extrusion through membrane filters can be performed with the help of centrifugation. 3. Microfluid cell fragmentation methods can be implemented to produce ACDVs. 4. Nitrogen cavitation results in cell disruption and ACDV formation. 5. Actin polymerization inhibitors and sulfhydryl blocking agents disrupt the cellular cytoskeleton and result in the formation of blebs and ultimately, ACDVs. Bottom-up strategy: (**A**) during a process of thin-film hydration, lipids are dissolved in organic solvent, which is evaporated to form a thin film. The film is rehydrated with an aqueous solution containing isolated cell membrane proteins and cargo. Agitation, such as stirring, results in multi-lamellar vesicle (MLV) and large unilamellar vesicle (LUV) formation. Extraction of the vesicles results in downsizing and the formation of small unilamellar vesicles (SUVs). (**B**) Biomimetic liposomes can also be formed by the microfluidic mixing of lipids in a water-miscible solution and cellular proteins and cargo in an aqueous solution.

**Figure 2 pharmaceutics-16-01306-f002:**
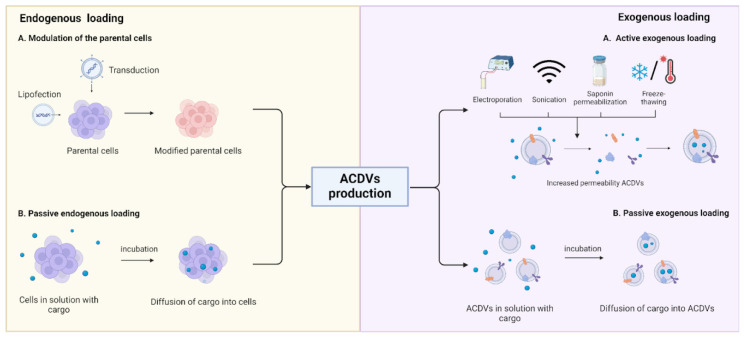
Cargo loading into ACDVs. Endogenous loading: (**A**) modification of parental cells by transfection or transduction to accumulate proteins or RNA of interest in the parental cells. During ACDV production, accumulated cargo is encapsulated inside the vesicles. (**B**) Similarly, the incubation of parental cells with cargo allows for the accumulation of cargo inside the cells that end up in the ACDVs after the production process. Exogenous loading: (**A**) active exogenous loading is based on transient pore formation in ACDVs with the help of electroporation, sonication, saponin, or freeze–thaw. Cargo in the solution can then enter the vesicles through pores and become encapsulated upon membrane resealing. (**B**) Passive cargo loading can be performed by the incubation of cargo with vesicles.

**Figure 3 pharmaceutics-16-01306-f003:**
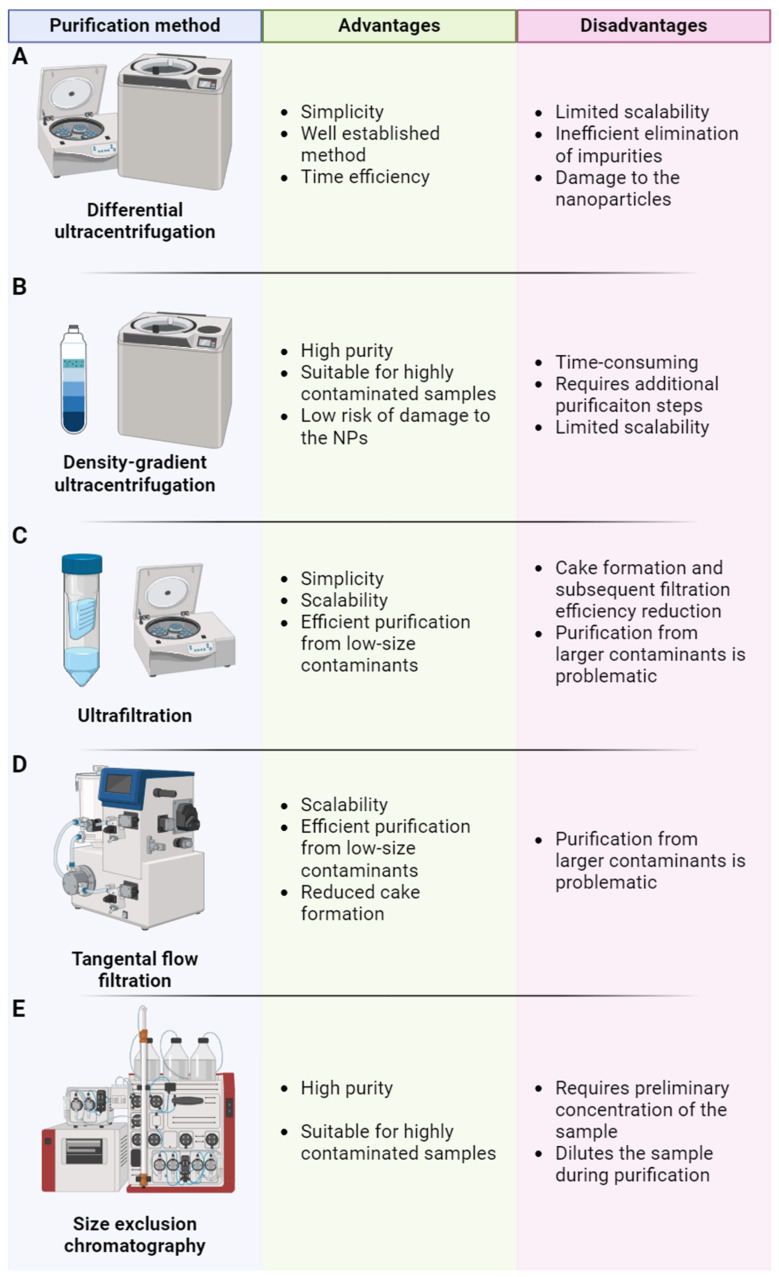
Summary of BMNP purification methods: (**A**) Differential ultracentrifugation, (**B**) Density-gradient ultracentrifugation, (**C**) Ultrafiltration, (**D**) Tangental flow filtration, (**E**) Size exclusion chromatography.

**Figure 4 pharmaceutics-16-01306-f004:**
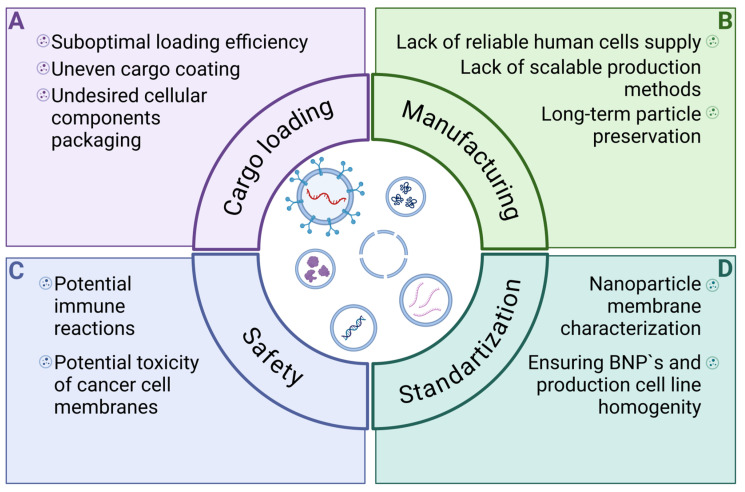
Technological challenges of BMNPs. (**A**) In the context of cargo loading, BMNPs exhibit suboptimal cargo loading efficiency. Additionally, the cargo may be unevenly distributed within the membrane, thereby not fully utilizing the capabilities of the delivery platform. Furthermore, undesirable cellular components can be incorporated into BMNPs, leading to reduced capacity and an increased risk of toxicity and adverse events. (**B**) A lack of cell supply can be detrimental to primary cell membrane-based BMNPs. The majority of currently utilized methods for BMNP preparation are characterized by poor scalability potential. Beyond manufacturing, long-term preservation of the final product remains a challenge. (**C**) While generally considered safe, a thorough investigation of potential immune responses and toxicity should be conducted for each type of BMNP. (**D**) Due to the abundance of potential cell membrane sources and variability in cell membranes during the cellular life cycle, the continuous monitoring of critical parameters at each manufacturing step is necessary to ensure consistency and minimize batch-to-batch variability.

**Table 1 pharmaceutics-16-01306-t001:** Comparison of technological aspects for producing natural EVs and BMNPs.

Features	Extracellular Vesicles (EVs)	Biomimetic Nanoparticles (BMNPs)			
		Cell-Membrane Coated Nanoparticles (CMCNPs)	Nanoghosts	Artificial Cell-Derived Vesicles (ACDVs)	Bottom-Up Strategies
Time-saving production	+	++	+++	+++	++
Lower manufacturing costs	+	++	++	+++	++
Homogeneity	+	++	++	+++	+++
Complexity	+++	++	++	+++	+
Circulation time	++	+/++	++	++	+
Targeted cargo delivery	+++	+/++	++	+++	+
Safety	+++	++	++	++	+

The symbols “+”represent the lowest quality, while “++++”indicate the highest quality.

**Table 2 pharmaceutics-16-01306-t002:** Feasibility of different types of cargo for packaging into BMNPs.

BMNP Type	Cargo Loading Feasibility					
	Nanoparticles	RNA	DNA	Proteins	Hydrophilic Small Mol	Hydrophobic Small Mol
CMCNP	+++ (not larger than 100 nm)	++ (with NPs)	++ (with NPs)	+ (With NPs)	++ (with NPs)	++ (with NPs)
Nanoghosts	+	++/+++	++	+++	+++	+
ACDVs	+	++/+++	++	++	++/+++	+
Biomimetic liposomes	+	++	+/++	+	++	+++
Fully synthetic vesicles	+	++	+/++	+	++	+++

The symbols “+”represent the lowest quality, while “++++”indicate the highest quality.

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
