# Peer review of "Biomimetic Nanoparticles for Basic Drug Delivery"

_pharmaceutics, 2024, doi:10.3390/pharmaceutics16101306_

Round 1

Reviewer 1 Report

Comments and Suggestions for Authors

The article provides an excellent and detailed overview of the promising field of biomimetic nanoparticles (BMNPs), highlighting their importance as innovative drug delivery vehicles. What stands out most convincingly is the ability of BMNPs to replicate the properties of natural extracellular vesicles, thereby enhancing their effectiveness in crossing biological barriers while ensuring high biocompatibility and minimal to no toxicity.

The in-depth analysis of various preparation strategies, including the use of modified biological membranes, demonstrates a solid and up-to-date knowledge of the subject, making this article a significant contribution to the field of drug delivery technologies. The inclusion of cell membrane-coated nanoparticles (CMCNPs), artificial cell-derived vesicles (ACDVs), and fully synthetic vesicles (fSVs) offers a comprehensive overview of the available technological options, making this work relevant to both academic research and clinical applications.

The discussion of "top-down" and "bottom-up" approaches, with a focus on cutting-edge techniques such as microfluidic fabrication, is particularly commendable, showcasing a clear understanding of the technological challenges and potential solutions for large-scale BMNP production. Overall, the article serves as a valuable reference for anyone interested in the advancement of nanotechnologies and their therapeutic applications, opening new horizons for the future of medicine.

In my opinion, the articles can be accepted in its current form.

Author Response

We sincerely thank the Reviewer for the exhaustive evaluation of our manuscript and Their positive assessment of its quality.

Reviewer 2 Report

Comments and Suggestions for Authors

The main objective of the work is giving a review regarding both "top-down" and "bottom-up" approaches for nanoparticle preparation, with a particular focus on techniques such as cell membrane coating, cargo loading, and microfluidic fabrication.

Furthermore, the authors critically address the technological challenges and potential solutions associated with the large-scale production and clinical application of biomimetic nannoparticles and related technologies.

In my opinion the review is comprehensive, well designed and exhaustive; it could represent an interesting support to the MISEV review, covering the missing field of "preparation/synthesis".

I would also like to report my appreciation regarding the quality and the clarity of the graphical abstract, of the figures and of the tables reported in the paper.

Author Response

We highly appreciate the Reviewer’s effort in reviewing our paper. It is very pleasant for the authors to receive such a positive evaluation of their work.

Reviewer 3 Report

Comments and Suggestions for Authors

In this review, the authors provide a thorough overview grounded in the fundamentals of biomimetic technologies, offering both a comprehensive perspective and an analytical discussion on the preparation and functionalization of biomimetic nanoparticles (BMNPs). The review highlights key technological challenges and potential solutions associated with the large-scale production and clinical application of BMNPs and related technologies. While the review is generally comprehensive, there are notable gaps in content, particularly in the classification of biomimetic nanoparticles. For instance, the direct fusion of extracellular vesicles (EVs) with liposomes or lipid nanoparticles is not adequately categorized or discussed. Therefore, I recommend major revisions

Specific comments are as follows:  

1.     Introduction: Please expand on the limitations and emphasize the need for developing BMNPs. For example, low yield and heterogeneity are mentioned but could benefit from further elaboration. Additionally, low cargo-loading efficiency should be discussed in more detail.

2.     The section on cargo loading into BMNPs needs more detailed categorization, with particular attention to feasibility. For instance, the limitations of nanoparticle size, as well as hydrophilicity and hydrophobicity, should be discussed. A table summarizing these points could be helpful.

3.     The differences between BMNPs and EVs in terms of clinical translation require more discussion. Are there similar challenges? In what aspects do BMNPs have an advantage over EVs in clinical applications? Some recent natural EV related to their clinical translation paper should be included here for discussion, such as 10.1016/j.tibtech.2024.08.007; doi.org/10.1002/adhm.202301010, etc.

4.     Are there any ongoing clinical trials or clinically approved products related to BMNPs?

5.     The conclusions and perspectives section is weak and requires a more detailed summary. Additionally, potential solutions and future outlook should be discussed in greater depth.

Author Response

Response:

We sincerely thank the Reviewer for thorough and professional evaluation of our manuscript. The comments made by the Reviewer are invaluable for the improvement of the manuscript. In response to the comments we have implemented several corrections, which are described in detail below:

Reviwer 3 comment 1:     Introduction: Please expand on the limitations and emphasize the need for developing BMNPs. For example, low yield and heterogeneity are mentioned but could benefit from further elaboration. Additionally, low cargo-loading efficiency should be discussed in more detail.

Authors response: Thank you. We have expanded the discussion of EVs’ limitations, which can be solved through BMNPs development, as follows:

“Mesenchymal stem cells (MSCs) are considered one of the safest sources for obtaining extracellular vesicles (EVs). However, their standardization is inadequate, with signifi-cant variations arising from the source of isolation and the characteristics of the donor [6]. The most common method for obtaining EVs is differential ultracentrifugation of conditioned media [7]. This method presents challenges related to scalability and the maintenance of the final product's purity, which limits its suitability for industrial ap-plications. Furthermore, the cargo packaging methods employed in EVs allow for the effective and efficient encapsulation of proteins and nucleic acids. However, technical challenges remain in the packaging of virus-like particles, nanoscale particles such as upconversion nanoparticles, and ribonucleoprotein (RNP) complexes like CRISPR-Cas9 [8,9].”

Reviwer 3 comment 2:     The section on cargo loading into BMNPs needs more detailed categorization, with particular attention to feasibility. For instance, the limitations of nanoparticle size, as well as hydrophilicity and hydrophobicity, should be discussed. A table summarizing these points could be helpful.

Response: Thank you for this useful suggestion. We have expanded the discussion of limitations that persist in the field of cargo loading into BMNPs and have also added Table 2, which summarizes the abovementioned limitations:

Loading of therapeutic cargo into BMNPs has several limitations into each type of BMNPs due to differences in particles production protocols and structure. During CMCNPs preparation, different types of nanoparticles are used for coating by membrane. Conceptually, particles of any size can be used for CMCNP. At the same time, it has been demonstrated that the clearance of vesicles from the bloodstream is size-dependent, with particles larger than 200 nm being quickly sequestered by macrophages [165]. Thus, the recommended size of the delivery vehicle should not exceed 150 nm. Given that the thickness of the lipid bilayer is ~10 nm, the inner diameter of the luminal space does not exceed ~120-130 nm [166]. Consequently, the membrane coating of nanoparticles larger than 100 nm may negatively impact the pharmacokinetics of CMCNP.

Almost all types of BMNPs can be utilized for the delivery of cargo DNA, par-ticularly short RNA. In the case of CMCNPs, therapeutic nucleic acids can be adsorbed onto positively charged nanoparticles prior to membrane coating. For artificial ACDVs, nucleic acids can be incorporated using endogenous methods. In biomimetic liposomes and fully synthetic vesicle technologies, packaging can be achieved through the thin-film hydration of lipids in the presence of nucleic acid solutions [118,130]. It should be noted that the use of sonication during thin-film hydration can damage large molecules, such as plasmids. Consequently, this method may be less feasible in such cases or may require additional pre-complexation with polymers, such as PEI. Additionally, optimized elec-troporation protocols can be applied to nearly all types of BMNPs, including ACDVs, nanoghosts, biomimetic liposomes, and fully synthetic vesicles. However, it is important to consider the potential for vesicle aggregation and cargo damage during electro-poration.

Therapeutic proteins represent another significant cargo for delivery via bio-mimetic systems. Nanoghosts are specifically optimized for protein delivery, and efficient packaging can be achieved during the production of vesicles [41]. ACDVs and nanoghosts can be effectively loaded with proteins using electroporation; however, potential vesicle damage must be considered [41]. The packaging of proteins using bottom-up approaches presents technical challenges.

Another important cargo for targeted delivery is small molecules. The coating of nanoparticles loaded with hydrophilic and hydrophobic molecules can be utilized for small molecule delivery via CMCNPs [32]. Effective loading of small molecules into nanoghosts and ACDVs can be achieved through electroporation and other exogenous physical and chemical methods, including sonication, freeze-thaw cycles, and saponin permeabilization [74]. Hydrophobic drugs are incorporated into nanoghosts and ACDVs by dissolving them in the membrane; however, this process requires aggressive organic solvents that can damage the vesicle membranes. Consequently, most packaging proto-cols rely on the use of the salt form of nominally hydrophobic drugs in aqueous solutions (for example, doxorubicin hydrochloride). In contrast, the hydration of lipid films in bottom-up approaches is much more effective for packaging hydrophobic drugs, as the cargo compounds can be mixed with lipids during the lipid film preparation [117].. The relative feasibility of BMNP packaging for different types of cargo is summarized in Table 2.”

Table 2. Feasibility of different types of cargo for packaging into BMNPs

BMNP type

Cargo loading feasibility

Nanoparticles

RNA

DNA

Proteins

Hydrophilic small mol

Hydrophobic small mol

CMCNP

+++ (not larger than 100 nm)

++ (with NPs)

++ (with NPs)

+ (With NPs)

++ (with NPs)

++ (with NPs)

Nanoghosts

+

++/+++

++

+++

+++

+

ACDVs

+

++/+++

++

++

++/+++

+

Biomimetic liposomes

+

++

+/++

+

++

+++

Fully synthetic vesicles

+

++

+/++

+

++

+++

Reviewer 3 comment 3:     The differences between BMNPs and EVs in terms of clinical translation require more discussion. Are there similar challenges? In what aspects do BMNPs have an advantage over EVs in clinical applications? Some recent natural EV related to their clinical translation paper should be included here for discussion, such as 10.1016/j.tibtech.2024.08.007; doi.org/10.1002/adhm.202301010, etc.

Response: The new text is now added, focusing on the advantages and potential challenges of BMNPs in context of clinical translation. The suggested papers were of great help in writing the new section:

It seems reasonable to posit that the transition to clinical trials for BMNPs will encounter challenges similar to those faced with EVs [170,171]. These challenges include complex manufacturing and standardization processes that comply with GMP, a lack of large-scale production capabilities, insufficient yield, and difficulties in preservation [172]. A significant advantage of BMNPs is their potential for large-scale production, which is not feasible for natural EVs. Technological advancements, such as extrusion and microfluidics-based methods, facilitate the industrial-scale manufacture of BMNPs [27]. The standardization of BMNPs is superior to that of EVs, which often exhibit variability in size, composition, and content due to biological factors. The controlled production processes for BMNPs ensure uniformity in size, shape, and surface properties, which is essential for reproducibility in clinical applications.

The incorporation of functional elements, such as ligands and targeting proteins, enhances the ability of BMNPs to interact with specific cells or tissues. For example, using membranes derived from macrophages or red blood cells can improve targeting capabilities while evading immune clearance. In contrast to the complex and la-bor-intensive processes required for natural EV separation, the production of BMNPs offers a more cost-effective and time-efficient approach [10].

However, one potential risk associated with BMNPs is the possibility of contami-nation with cellular contents that may trigger immune responses [43]. The incorporation of cellular debris or unwanted proteins from the donor cell line into BMNPs could in-duce inflammation upon introduction into the body. Therefore, it is essential to imple-ment rigorous purification and quality control processes to mitigate this risk. The like-lihood of an immune response being elicited is contingent upon the source of the cell membranes used for coating. This risk is particularly pronounced when the membranes contain MHC molecules or other immunogenic components [167].

Additionally, the processes employed to prepare BMNPs, including sonication and homogenization, may damage the therapeutic cargo, potentially reducing its efficacy [118,130]. Incomplete coating of nanoparticles with cell membranes may occur, leading to diminished functionality and efficiency in targeted delivery. Such incomplete cover-age could compromise the biocompatibility and targeting capabilities of BMNPs [51].”

Reviewer 3 comment 4:     Are there any ongoing clinical trials or clinically approved products related to BMNPs?

Response: We appreciate the comment. We have mentioned in our manuscript that BMNPs are still yet to be applied clinically (“The use of BMNPs is still in its infancy, as they have not been applied clinically. Clinicals trials of BMNPs are more difficult than those of traditional small molecular drugs”). However, to the best of the authors’ knowledge, there is no ongoing trials nor clinically approved products.

Reviewer 3 comment 5:    The conclusions and perspectives section is weak and requires a more detailed summary. Additionally, potential solutions and future outlook should be discussed in greater depth.

Response: We have reworked and expanded the “Conclusions and Perspectives” section as follows:

EVs play a crucial role in cell-to-cell communication by transporting nucleic acids, proteins, and various types of cargo. Since their discovery, EVs have been recognized as promising delivery carriers capable of encapsulating diverse therapeutic agents. Nu-merous clinical trials are currently underway to explore a wide range of therapeutic applications. However, several limitations hinder the clinical use of EVs, including low yields, high heterogeneity, and challenges in standardization. These factors complicate the approval process for medical applications involving EVs.

To address these issues, the production of BMNPs has emerged as a viable solution, combining the high bioavailability of natural vesicles with more efficient and cost-effective manufacturing processes. Various manufacturing approaches for BMNPs have been developed, including CMCNPs, nanoghosts, artificial cell-derived vesicles ACDVs, and liposomes integrated with membrane proteins. Establishing standardized cell sources for optimal biomimetic production is essential for advancing BMNP tech-nologies. Consistent and efficient cell lines are necessary to ensure uniform BMNP production.

These standardized cell lines can be further modified to incorporate specific surface elements, such as targeting proteins, "don’t eat me" signals, or ligands for surface re-ceptors, particularly those that require a membrane-bound state for functionality. Ad-ditionally, developing cell lines that lack MHC-I molecules can help mitigate immune responses, enhancing the safety of BMNPs for therapeutic use.

Another significant advantage of BMNPs is their ability to encapsulate content that is either inaccessible or ineffective when delivered via exosomes. BMNPs can accom-modate a wide range of cargo, including nanoparticles, viruses, and gene-editing tools such as CRISPR and RNA interference (RNAi). BMNPs exhibit distribution kinetics comparable to those of natural EVs and often outperform liposomes in various contexts.

In terms of production, BMNPs offer substantial advantages over exosomes in cost, time efficiency, and yield. The production processes for BMNPs are faster, more scalable, and less expensive, making them more suitable for large-scale therapeutic applications. Furthermore, BMNP production can be precisely controlled, resulting in nanoparticles with consistent size and composition—critical factors for therapeutic efficacy and regu-latory approval.

Enhancements in purification and isolation methods are necessary to improve the scalability and purity of BMNPs. Advanced techniques such as chromatography, tan-gential flow filtration, and affinity-based methods could effectively remove unwanted biological components without compromising the integrity of the nanoparticles.

Moving forward, the development of BMNPs must focus on optimizing production processes, improving cargo-loading efficiency, and enhancing vesicle surface function-alization. Addressing these challenges is essential for creating next-generation drug de-livery platforms.”

Reviewer 4 Report

Comments and Suggestions for Authors

The manuscript reports on a review of biomimetic nanoparticles, specifically, on cell membrane-coated nanoparticles (CMCNPs), artificial cell-derived vesicles (ACDVs), and fully synthetic vesicles (fSVs). After the introduction, the review starts by discussing top-down approaches for the synthesis of the biomimetic nanoparticles, their loading and the materials involved. A similar approach is used on the subsequent discussion of the bottom-up synthesis methods. Their purification is reviewed. The manuscript then continues with the discussion of constraints and challenges, of the pharmokinetics and drug delivery and of the conclusions and perspectives. The manuscript presents an original and detailed review, with abundant references for the interested reader, and only needs minor revisions. I have the following questions.

- The manuscript has very few figures (only 2 n its 27 pages). It would benefit from having more images to illustrate the points.

- The manuscript uses lots of acronyms and needs to be careful for consistency. For example, the acronym for biomimetic nanoparticles is sometimes written as BMNPs and other times is BNPs. Other acronyms should also be checked.

- The authors use curved parentheses in the citations, which is not standard and is confusion (it seems numbering values, not cited references numbers). The authors should use straight parentheses for the citations, as this is the standard practice.

- All the acronyms should be defined on their first used. For example, MISEV2023 is used, but never defined. The others should be checked.

- Frequently the author reference the name of a researcher and but do not put citation to their work. For example, on page 9 it is written “In experiment by Kaneti et al….”, but there is no citation on what he has done. Does it appear much later ? It should be put immediately after the “et al” near the name. The same happens on page 9 with Toledano Furman et al., on page 10 with Oieni et al. and in other places in the manuscript. The citations to their work should be near their names.

Author Response

We sincerely thank the Reviewer for their time and effort. We have corrected the manuscript in order to address all of the comments made by the reviewer. The abovementioned corrections are discussed below:

Reviwer 4 comment 1: The manuscript has very few figures (only 2 n its 27 pages). It would benefit from having more images to illustrate the points.

Answer: Thank you for the comment. We have added two more figures (Figure 3 and Figure 4). Other comments were addressed by correction of different parts of the manuscript.

Round 2

Reviewer 3 Report

Comments and Suggestions for Authors

The authors have satisfactorily addressed all of my concerns. Excellent work!